# DUAL DIFFUSION IMPLICIT BRIDGES FOR IMAGE-TO-IMAGE TRANSLATION

**Xuan Su**[1]    **Jiaming Song**[2]    **Chenlin Meng**[1]    **Stefano Ermon**[1,3]
[1]Stanford University    [2]NVIDIA    [3]CZ Biohub
{suxuan,chenlin,ermon}@cs.stanford.edu, jiamings@nvidia.com

## ABSTRACT

Common image-to-image translation methods rely on joint training over data from both source and target domains. The training process requires concurrent access to both datasets, which hinders data separation and privacy protection; and existing models cannot be easily adapted for translation of new domain pairs. We present Dual Diffusion Implicit Bridges (DDIBs), an image translation method based on diffusion models, that circumvents training on domain pairs. Image translation with DDIBs relies on two diffusion models trained independently on each domain, and is a two-step process: DDIBs first obtain latent encodings for source images with the source diffusion model, and then decode such encodings using the target model to construct target images. Both steps are defined via ordinary differential equations (ODEs), thus the process is cycle consistent only up to discretization errors of the ODE solvers. Theoretically, we interpret DDIBs as concatenation of source to latent, and latent to target Schrödinger Bridges, a form of entropy-regularized optimal transport, to explain the efficacy of the method. Experimentally, we apply DDIBs on synthetic and high-resolution image datasets, to demonstrate their utility in a wide variety of translation tasks and their inherent optimal transport properties.

## 1 INTRODUCTION

Transferring images from one domain to another while preserving the content representation is an important problem in computer vision, with wide applications that span style transfer (Xu et al., 2021; Sinha et al., 2021) and semantic segmentation (Li et al., 2020). In tasks such as style transfer, it is usually difficult to obtain paired images of realistic scenes and their artistic renditions. Consequently, unpaired translation methods are particularly relevant, since only the datasets, and not the one-to-one correspondence between image translation pairs, are required. Common methods on unpaired translation are based on generative adversarial networks (GANs, Goodfellow et al. (2014); Zhu et al. (2017)) or normalizing flows (Grover et al., 2020). Training such models typically involves minimizing an adversarial loss between a specific pair of source and target datasets.

While capable of producing high-quality images, these methods suffer from a severe drawback in their *adaptability* to alternative domains. Concretely, a translation model on a source-target pair is trained specifically for this domain pair. Provided a different pair, existing, bespoke models cannot be easily adapted for translation. If we were to do pairwise translation among a set of domains, the total number of models needed is quadratic in the number of domains – an unacceptable computational cost in practice. One alternative is to find a shared domain that connects to each source / target domains as in StarGANs (Choi et al., 2018). However, the shared domain needs to be carefully chosen *a priori*; if the shared domain contains less information than the target domain (*e.g.* sketches v.s. photos), then it creates an unwanted information bottleneck between the source and target domains.

An additional disadvantage of existing models resides in their *lack of privacy protection* of the datasets: training a translation model requires access to both datasets simultaneously. Such setting may be inconvenient or impossible, when data providers are reluctant about giving away their data; or for certain privacy-sensitive applications such as medical imaging. For example, quotidian hospital usage may require translation of patients' X-ray and MRI images taken from machines in other hospitals. Most existing methods will fail in such scenarios, as joint training requires aggregating confidential imaging data across hospitals, which may violate patients' privacy.

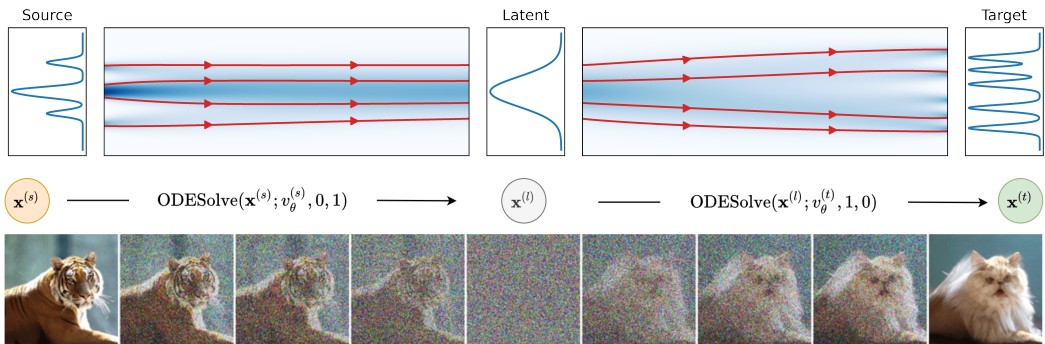

Figure 1: **Dual Diffusion Implicit Bridges**: DDIBs leverage two ODEs for image translation. Given a source image $\mathbf{x}^{(s)}$, the source ODE runs in the forward direction to convert it to the latent $\mathbf{x}^{(l)}$, while the target, reverse ODE then constructs the target image $\mathbf{x}^{(t)}$. (*Top*) Illustration of the DDIBs idea between two one-dimensional distributions. (*Bottom*) DDIBs from a tiger to a cat using a pretrained conditional diffusion model.

In this paper, we seek to mitigate both problems of existing image translation methods. We present Dual Diffusion Implicit Bridges (DDIBs), an image-to-image translation method inspired by recent advances in diffusion models (Song et al., 2020a;b), that decouples paired training, and empowers the domain-specific diffusion models to stay applicable in other pairs wherever the domain appears again as the source or the target. Since the training process now concentrates on one dataset at a time, DDIBs can also be applied in federated settings, and not assume access to both datasets during model training. As a result, owners of domain data can effectively preserve their data privacy.

Specifically, DDIBs are developed based on the method known as denoising diffusion implicit models (DDIMs, Song et al. (2020a)). DDIMs invent a particular parameterization of the diffusion process, that creates a smooth, deterministic and reversible mapping between images and their latent representations. This mapping is captured using the solution to a so-called *probability flow* (PF) ordinary differential equation (ODE) that forms the cornerstone of DDIBs. Translation with DDIBs on a source-target pair requires two different PF ODEs: the source PF ODE converts input images to the latent space; while the target ODE then synthesizes images in the target domain.

Crucially, trained diffusion models are specific to the individual domains, and rely on no domain pairing information. Effectively, DDIBs make it possible to save a trained model of a certain domain for future use, when it arises as the source or target in a new pair. Pairwise translation with DDIBs requires only a *linear* number of diffusion models (which can be further reduced with conditional models (Dhariwal & Nichol, 2021)), and training does not require scanning both datasets concurrently.

Theoretically, we analyze the DDIBs translation process to highlight two important theoretical properties. First, the probability flow ODEs in DDIBs, in essence, comprise the solution of a special Schrödinger Bridge Problem (SBP) with linear or degenerate drift (Chen et al., 2021a), between the data and the latent distributions. This justification of DDIBs from an optimal transport viewpoint that alternative translation methods lack serves as a theoretical advantage of our method, as DDIBs are the most OT-efficient translation procedure while alternate methods may not be. Second, DDIBs guarantee exact cycle consistency: translating an image to and back from the target space reinstates the original image, only up to discretization errors introduced in the ODE solvers.

Experimentally, we first present synthetic experiments on two-dimensional datasets to demonstrate DDIBs' cycle-consistency property. We then evaluate our method on a variety of image modalities, with qualitative and quantitative results: we validate its usage in example-guided color transfer, paired image translation, and conditional ImageNet translation. These results establish DDIBs as a scalable, theoretically rigorous addition to the family of unpaired image translation methods.

## 2 Preliminaries

### 2.1 Score-based Generative Models (SGMs)

While our actual implementation utilizes DDIMs, we first briefly introduce the broader family of models known as score-based generative models. Two representative models of this family are *score matching with Langevin dynamics* (SMLD) (Song & Ermon, 2019) and *denoising diffusion*

*probabilistic models* (DDPMs) (Ho et al., 2020). Both methods are contained within the framework of Stochastic Differential Equations (SDEs) proposed in Song et al. (2020b).

**Stochastic Differential Equation (SDE) Representation**   Song et al. (2020b); Anderson (1982) use a forward and a corresponding backward SDE to describe general diffusion and the reversed, generative processes:

$$\mathrm{d}\mathbf{x} = \mathbf{f}(\mathbf{x}, t)\,\mathrm{d}t + g(t)\,\mathrm{d}\mathbf{w}\,, \quad \mathrm{d}\mathbf{x} = [\mathbf{f} - g^2 \nabla_{\mathbf{x}} \log p_t(\mathbf{x})]\,\mathrm{d}t + g(t)\,\mathrm{d}\mathbf{w} \tag{1}$$

where $\mathbf{w}$ is the standard Wiener process, $\mathbf{f}(\mathbf{x}, t)$ is the vector-valued drift coefficient, $g(t)$ is the scalar diffusion coefficient, and $\nabla_{\mathbf{x}} \log p_t(\mathbf{x})$ is the score function of the noise perturbed data distribution (as defined by the forward SDE with initial condition $p_0(\mathbf{x})$ being the data distribution). At the endpoints $t = \{0, 1\}$, the forward Eq. (1) admits the data distribution $p_0$ and the easy-to-sample prior $p_1$ as the boundary distributions. Within this framework, the SMLD method can be described using a *Variance-Exploding* (VE) SDE with increasing noise scales $\sigma(t)$: $\mathrm{d}\mathbf{x} = \sqrt{\mathrm{d}[\sigma^2(t)]\,/\,\mathrm{d}t}\,\mathrm{d}\mathbf{w}$. In comparison, DDPMs are endowed with a *Variance-Preserving* (VP) SDE: $\mathrm{d}\mathbf{x} = -[\beta(t)/2]\mathbf{x}\,\mathrm{d}t + \sqrt{\beta(t)}\,\mathrm{d}\mathbf{w}$ with $\beta(t)$ being another noise sequence. Notably, the VP SDE can be reparameterized into an equivalent VE SDE (Song et al., 2020a).

**Probability Flow ODE**   Any diffusion process can be represented by a *deterministic* ODE that carries the same marginal densities as the diffusion process throughout its trajectory. This ODE is termed the *probability flow* (PF) ODE (Song et al., 2020b). PF ODEs enable *uniquely identifiable encodings* (Song et al., 2020b) of data, and are central to DDIBs as we solve these ODEs for forward and reverse conversion between data and their latents. For the forward SDE introduced in Eq. (1), the equivalent PF ODE holds the following form:

$$\mathrm{d}\mathbf{x} = \left[\mathbf{f}(\mathbf{x}, t) - \frac{1}{2}g(t)^2 \nabla_{\mathbf{x}} \log p_t(\mathbf{x})\right]\mathrm{d}t \tag{2}$$

which follows immediately from the SDEs given the score function. In practice, we use $\theta$-parameterized score networks $\mathbf{s}_{t,\theta} \approx \nabla_{\mathbf{x}} \log p_t(\mathbf{x})$ to approximate the score function. Training such networks relies on a variational lower bound, described in Ho et al. (2020) and in Appendix B. We may then employ numerical ODE solvers to solve the above ODE and construct $\mathbf{x}$ at different times. Empirically, it has been demonstrated that SGMs have relatively low discretization errors when reconstructing $\mathbf{x}$ at $t = 0$ via ODE solvers (Song et al., 2020a). For conciseness, we use $v_\theta = \mathrm{d}\mathbf{x}\,/\,\mathrm{d}t$ to denote the $\theta$-parameterized velocity field (as defined from Eq. (2), where we replace $\nabla_{\mathbf{x}} \log p_t(\mathbf{x})$ with $\mathbf{s}_{t,\theta}$), and use the symbol ODESolve to denote the mapping from $\mathbf{x}(t_0)$ to $\mathbf{x}(t_1)$:

$$\text{ODESolve}(\mathbf{x}(t_0); v_\theta, t_0, t_1) = \mathbf{x}(t_0) + \int_{t_0}^{t_1} v_\theta(t, \mathbf{x}(t))\,\mathrm{d}t\,, \tag{3}$$

which allows us to abstract away the exact model (be it a score-based or a diffusion model), or the integrator used. In our experiments, we implement the ODE solver in DDIMs (Song et al., 2020a) (Appendix B); while we acknowledge other available ODE solvers that are usable within our framework, such as the DPM-solver (Lu et al., 2022), the Exponential Integrator (Zhang & Chen, 2022), and the second-order Heun solver (Karras et al., 2022).

## 2.2   SCHRÖDINGER BRIDGE PROBLEM (SBP)

Our analysis shows that DDIBs are Schrödinger Bridges (Chen et al., 2016; Léonard, 2013) between distributions. Let $\Omega = C([0, 1]; \mathbb{R}^n)$ be the path space of $\mathbb{R}^n$-valued continuous functions over the time interval $[0, 1]$; and $\mathcal{D}(p_0, p_1)$ be the set of distributions over $\Omega$, with marginals $p_0, p_1$ at time $t = 0, t = 1$, respectively. Given a prior reference measure $W^1$, the well-known *Schrödinger Bridge Problem* (SBP) seeks the most probable evolution across time $t$ between the marginals $p_0$ and $p_1$:

**Problem 1** (Schrödinger Bridge Problem). *With prescribed distributions $p_0, p_1$ and a reference measure $W$ as the prior, the SBP finds a distribution from $\mathcal{D}(p_0, p_1)$ that minimizes its KL-divergence to $W$: $P_{SBP} \coloneqq \arg\min\{D_{KL}(P\|W) \mid P \in \mathcal{D}(p_0, p_1)\}$.*

---

[1]In our application, the reference measure is set to the measure of Eq. (1), as per Chen et al. (2021a).

---

**Algorithm 1** High-level Pseudo-code for DDIBs

---

**Input:** data sample from source domain $\mathbf{x}^{(s)} \sim p_s(\mathbf{x})$, source model $v_\theta^{(s)}$, target model $v_\theta^{(t)}$.
**Output:** $\mathbf{x}^{(t)}$, the result in the target domain.
$\mathbf{x}^{(l)} = \text{ODESolve}(\mathbf{x}^{(s)}; v_\theta^{(s)}, 0, 1)$      // obtain latent code from source domain data
$\mathbf{x}^{(t)} = \text{ODESolve}(\mathbf{x}^{(l)}; v_\theta^{(t)}, 1, 0)$      // obtain target domain data from latent code
**return** $\mathbf{x}^{(t)}$

---

The minimizer, $P_{\text{SBP}}$, is dubbed the *Schrödinger Bridge* between $p_0$ and $p_1$ over prior $W$. The SBP has connections to the Monge-Kantorovich (MK) optimal transport problem (Chen et al., 2021b). While the basic MK problem seeks the cost-minimizing plan to transport masses between distributions, the SBP incorporates an additional entropy term (for details, see Page 61 of Peyré et al. (2019)) .

**Relationship Between SBPs and SGMs**    Chen et al. (2021a) establishes connections between SGMs and SBPs. In summary, SGMs are implicit optimal transport models, corresponding to SBPs with linear or degenerate drifts. General SBPs additionally accept fully nonlinear diffusion. To formalize this observation, the authors first establish similar forward and backward SDEs for SBPs:

$$\mathrm{d}\mathbf{x} = [\mathbf{f} + g^2 \nabla_\mathbf{x} \log \Phi_t(\mathbf{x})]\,\mathrm{d}t + g(t)\,\mathrm{d}\mathbf{w}\,, \quad \mathrm{d}\mathbf{x} = [\mathbf{f} - g^2 \nabla_\mathbf{x} \log \hat{\Phi}_t(\mathbf{x})]\,\mathrm{d}t + g(t)\,\mathrm{d}\mathbf{w} \quad (4)$$

where $\Phi, \hat{\Phi}$ are the *Schrödinger factors* that satisfy density factorization: $p_t(\mathbf{x}) = \Phi_t(\mathbf{x})\hat{\Phi}_t(\mathbf{x})$. The vector-valued quantities $\mathbf{z}_t = g(t)\nabla_\mathbf{x} \log \Phi_t(\mathbf{x}), \hat{\mathbf{z}}_t = g(t)\nabla_\mathbf{x} \log \hat{\Phi}_t(\mathbf{x})$ fully characterize dynamics of the SBP, thus can be considered as the forward, backward "policies", analogous to policy-based methods described in Schulman et al. (2015); Pereira et al. (2019). To draw a link between SBPs and SGMs, the data log-likelihood objective for SBPs is computed and shown to be equal to that of SGMs with special choices of $\mathbf{z}_t, \hat{\mathbf{z}}_t$ (derivation details in Chen et al. (2021a)). Importantly, likelihood equality occurs with the following policies:

$$(\mathbf{z}_t, \hat{\mathbf{z}}_t) = (0, g(t)\,\nabla_\mathbf{x} \log p_t(\mathbf{x})) \quad (5)$$

When the marginal $p_1$ at time $t = 1$ is equal to the prior distribution, it is known that such $(\mathbf{z}_t, \hat{\mathbf{z}}_t)$ are achieved. Since in SGMs, the end marginal $p_1$ is indeed the standard Gaussian prior, their log-likelihood is equivalent to that of SBPs. This suggests that SGMs are a special case of SBPs with degenerate forward policy $\mathbf{z}_t$ and a multiple of the score function as its backward $\hat{\mathbf{z}}_t$.

**Probability Flow ODE**    In a similar vein to the SGM SDEs, a deterministic PF ODE can be derived for SBPs with identical marginal densities across $t \in [0, 1]$. The following PF ODE specifies the probability flow of the optimal processes of the SBP defined in Eq. (4) (Chen et al., 2021a):

$$\mathrm{d}\mathbf{x} = \left[\mathbf{f}(\mathbf{x}, t) + g(t)\,\mathbf{z} - \frac{1}{2}g(t)(\mathbf{z} + \hat{\mathbf{z}})\right]\mathrm{d}t \quad (6)$$

where $\mathbf{z}$ depends on $\mathbf{x}$. We shall show that the PF ODEs for SGMs and SBPs are equivalent. Thus, flowing through the PF ODEs in DDIBs is equivalent to flowing through special Schrödinger Bridges, with one of the marginals being Gaussian.

## 3    DUAL DIFFUSION IMPLICIT BRIDGES

DDIBs leverage the connections between SGMs and SBPs to perform image-to-image translation, with two diffusion models trained separately on the two domains. DDIBs contain two steps, described in Alg. 1 and illustrated in Fig. 1. At the core of the algorithm is the ODE solver ODESolve from Eq. (3). Given a source model represented as a vector field, *i.e.*, $v_\theta^{(s)}$ defined from Eq. (2), DDIBs first apply ODESolve in the source domain to obtain the encoding $\mathbf{x}^{(s)}$ of the image at the end time $t = 1$; we refer to this as the *latent code* (associated with the diffusion model for the domain). Then, the source latent code is fed as the initial condition (target latent code at $t = 1$) to ODESolve with the target model $v_\theta^{(t)}$ to obtain the target image $\mathbf{x}^{(t)}$. As discussed earlier, we implement ODESolve with DDIMs (Song et al., 2020a), which are known to have reasonably small discretization errors. While

recent developments in higher order ODE solvers (Zhang & Chen, 2022; Lu et al., 2022; Karras et al., 2022) that generalize DDIMs can also be used here, we leave this investigation to future work.

Despite the simplicity of the method, DDIBs have several advantages over prior methods, which we discuss below.

**Exact Cycle Consistency**    A desirable feature of image translation algorithms is the *cycle consistency* property: transforming a data point from the source domain to the target domain, and then back to source, will recover the original data point in the source domain. The following proposition validates the cycle consistency of DDIBs.

**Proposition 3.1** (DDIBs Enforce Exact Cycle Consistency). *Given a sample from source domain* $\mathbf{x}^{(s)}$, *a source diffusion model* $v_\theta^{(s)}$, *and a target model* $v_\theta^{(t)}$, *define:*

$$\mathbf{x}^{(l)} = \text{ODESolve}(\mathbf{x}^{(s)}; v_\theta^{(s)}, 0, 1); \quad \mathbf{x}^{(t)} = \text{ODESolve}(\mathbf{x}^{(l)}; v_\theta^{(t)}, 1, 0); \tag{7}$$

$$\mathbf{x}'^{(l)} = \text{ODESolve}(\mathbf{x}^{(t)}; v_\theta^{(t)}, 0, 1); \quad \mathbf{x}'^{(s)} = \text{ODESolve}(\mathbf{x}'^{(l)}; v_\theta^{(s)}, 1, 0) \tag{8}$$

*Assume zero discretization error. Then,* $\mathbf{x}^{(s)} = \mathbf{x}'^{(s)}$.

As PF ODEs are used, the cycle consistency property is guaranteed. In practice, even with discretization error, DDIBs incur almost negligible cycle inconsistency (Section 4.1). In contrast, GAN-based methods are not guaranteed the cycle consistency property by default, and have to incorporate additional training terms to optimize for cycle consistency over two domains.

**Data Privacy in Both Domains**    In the DDIBs translation process, only the source and target diffusion models are required, whose training processes do not depend on knowledge of the domain pair *a priori*. In fact, this process can even be performed in a privacy sensitive manner (graphic illustration in Appendix A). Let Alice and Bob be the data owners of the source and target domains, respectively. Suppose Alice intends to translate images to the target domain. However, Alice does not want to share the data with Bob (and vice versa, Bob does not want to release their data either). Then, Alice can simply train a diffusion model with the source data, encode the data to the latent space, transmit the latent codes to Bob, and next ask Bob to run their trained diffusion model and send the results back. In this procedure, only the latent code and the target results are transmitted between the two data vendors, and both parties have naturally ensured that their data are not directly revealed.

**DDIBs are Two Concatenated Schrödinger Bridges**    DDIBs link the source data distribution to the latent space, and then to the target distribution. What is the nature of such connections between distributions? We offer an answer from an optimal transport perspective: these connections are special *Schrödinger Bridges* between distributions. This, in turn, explicates the name of our method: dual diffusion implicit bridges are based on denoising *diffusion implicit* models (Song et al., 2020a), and consist of *two* separate Schrödinger *Bridges* that connect the data and latent distributions. Specifically, as considered earlier, when conditions about the policies $\mathbf{z}_t, \hat{\mathbf{z}}_t$ in Eq. (5) and the density $p_1(\mathbf{x})$ being a Gaussian prior are met, the data likelihoods (at $t = 0$) for SGMs and SBPs are identical. Indeed, these conditions are fulfilled in SGMs and particularly in DDIMs. This verifies SGMs as special linear or degenerate SBPs. Forward and reverse solving the PF ODE for SGMs, as done in DDIBs, is equivalent to flowing through the optimal processes of particular SBPs:

**Proposition 3.2** (PF ODE Equivalence[2]). *Eq. (2) is equivalent to Eq. (6) with forward, backward policies* $(\mathbf{z}_t, \hat{\mathbf{z}}_t) = (0, g\nabla_\mathbf{x} \log p_t(\mathbf{x}))$ *as attained in SGMs and particularly in DDIMs.*

Thus, DDIBs are intrinsically entropy-regularized optimal transport: they are Schrödinger Bridges between the source and the latent, and between the latent and the target distributions. The translation process can then be recognized as traversing through two concatenated Schrödinger Bridges, one forward and one reversed. The mapping is unique and minimizes a (regularized) optimal transport objective, which probably elucidates the superior performance of DDIBs. In contrast, if we train the source and target models separately with normalizing flow models that are not inborn with such a connection, there are many viable invertible mappings, and the resulting image translation algorithm may not necessarily have good performance. This is probably the reason why AlignFlow (Grover et al., 2020) still has to incorporate an adversarial loss even when cycle-consistency is guaranteed.

---

[2]Proof in Appendix D.

Table 1: Cycle consistency of DDIBs. Experiment legend, PR ↻ PS, means that we translate from PR to PS and then back. The numbers are the averaged L2 distances between the original points and their coordinates after cycle translation. Data points are standardized to have unit variance.

| PR ↻ PS | PS ↻ CS | CR ↻ PR | CR ↻ CS | M ↻ CB |
|---------|---------|---------|---------|--------|
| 0.0143  | 0.0065  | 0.0106  | 0.0078  | 0.0122 |

## 4 EXPERIMENTS

We present a series of experiments to demonstrate the effectiveness of DDIBs. First, we describe synthetic experiments on two-dimensional datasets, to corroborate DDIBs' cycle-consistent and optimal transport properties. Next, we validate DDIBs on a variety of image translation tasks, including color transfer, paired translation, and conditional ImageNet translation. [3][4]

### 4.1 2D SYNTHETIC EXPERIMENTS

We first perform domain translation on synthetic datasets drawn from complex two-dimensional distributions, with various shapes and configurations, in Fig. 2a. In total, we consider six 2D datasets: Moons (M); Checkerboards (CB); Concentric Rings (CR); Concentric Squares (CS); Parallel Rings (PR); and Parallel Squares (PS). The datasets are all normalized to have zero mean, and identity covariance. We assign colors to points based on the point identities (*i.e.*, if a point in the source domain is red, its corresponding point in the target domain is also colored red). Clearly, the transformation is *smooth* between columns. For example, on the top-right corner, red points in the CR dataset are mapped to similar coordinates, both in the latent and in the target dimensions.

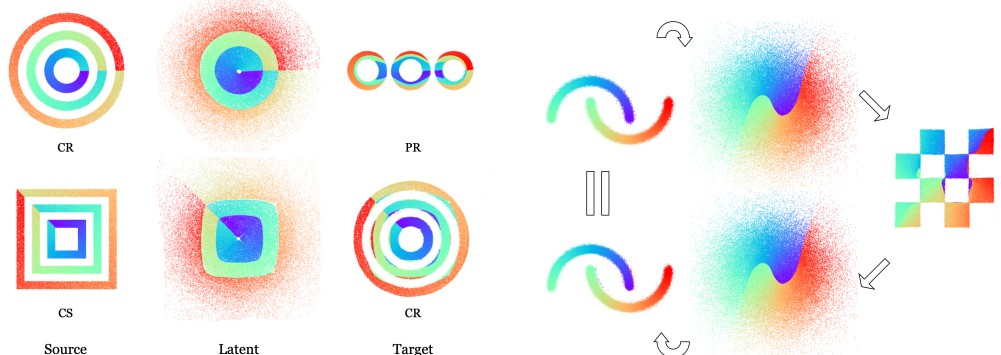

(a) Smooth translation of synthetic datasets. (*Left*) The source datasets: CR and CS. (*Middle*) DDIBs' latent code representation. (*Right*) Results of translation to the target domains.

(b) **Cycle consistency**: After translating the Moons dataset to Checkerboards and then back to Moons, DDIBs restore almost the exact same points as the original ones.

Figure 2: Smoothness and cycle consistency of DDIBs.

**Cycle Consistency**   Fig. 2b illustrates the cycle consistency property guaranteed by DDIBs. It concerns 2D datasets: Moons, and Checkerboards. Starting from the Moons dataset, DDIBs first obtain the latent codes and construct the Checkerboards points. Next, DDIBs do translations in the reverse direction, transforming the points back to the latent and the Moons space. After this round trip, points are approximately mapped to their original positions. A similar, smooth color topology is observed in this experiment. Table 1 reports quantitative evaluation results on cycle-consistent translation among multiple datasets. As the datasets are normalized to unit standard deviation, the reported values are negligibly small and endorse the cycle consistent property of DDIBs.

---

[3]**Project**: https://suxuann.github.io/ddib/
[4]**Code**: https://github.com/suxuann/ddib/

Table 2: Mean Squared Error (MSE) comparing color transfer results of DDIBs with common OT methods on two images. Each number represents the MSE between DDIBs and the corresponding OT method. MSE is computed pixel-wise after normalizing images to $[-1, 1]$.

| IMAGE | EMD | SINKHORN | LINEAR | GAUSSIAN |
|---|---|---|---|---|
| TARGET 1 | 0.0337 | 0.0281 | 0.0352 | 0.0370 |
| TARGET 2 | 0.0293 | 0.0326 | 0.0500 | 0.0751 |

Table 3: MSE comparing DDIBs and baselines on paired test sets. MSE is computed pixel-wise after normalizing images to $[-1, 1]$.

| DATASET | MODEL | A → B | B → A | DATASET | MODEL | A → B | B → A |
|---|---|---|---|---|---|---|---|
| | CYCLEGAN | 0.7129 | 0.3286 | | CYCLEGAN | 0.0245 | 0.0953 |
| FACADES | ALIGNFLOW | 0.5801 | **0.2512** | MAPS | ALIGNFLOW | 0.0209 | **0.0897** |
| | DDIBS | **0.5312** | 0.3946 | | DDIBS | **0.0194** | 0.1302 |

## 4.2 EXAMPLE-GUIDED COLOR TRANSFER

DDIBs can be used on an interesting application: example-guided color transfer. This refers to the task of modifying the colors of an input image, conditioned on the color palette of a reference image. To use DDIBs for color transfer, we train one diffusion model per image, on its normalized RGB space. During translation, DDIBs obtain encodings of the original colors, and apply the diffusion model of the reference image to attain the desired color palette. Fig. 3 visualizes our color experiments.

**Comparison to Alternative OT Methods**   As DDIBs are related to regularized OT, we compare the pixel-wise MSEs between color-transferred images generated by DDIBs, and images produced by alternate methods, in Table 2. We include four OT methods for comparison: Earth Mover's Distance; Sinkhorn distance (Cuturi, 2013); linear and Gaussian mapping estimation (Perrot et al., 2016). Results of DDIBs are very close to those of OT methods. Appendix E.2 details full color translation results.

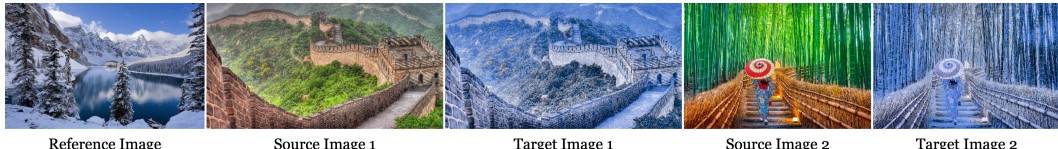

| Reference Image | Source Image 1 | Target Image 1 | Source Image 2 | Target Image 2 |

Figure 3: **Example-Guided Color Transfer**: Given the first image as the reference image, DDIBs modify the colors of two input images to similarly follow a snowy winter color palette.

## 4.3 QUANTITATIVE TRANSLATION EVALUATION

Quantitatively, we demonstrate that DDIBs deliver competitive results on paired domain tests. Such numerical evaluation is despite that DDIBs are formulated with a weaker setting: diffusion models are trained independently, on separate datasets. In comparison, methods such as CycleGAN and AlignFlow assume access to both datasets during training and jointly optimize for the translation loss.

**Paired Domain Translation**   As in similar works, we evaluate DDIBs on benchmark paired datasets (Zhu et al., 2017): Facades and Maps. Both are image segmentation tasks. In the pairs of datasets, one dataset contains real photos taken via a camera or a satellite; while the other comprises the corresponding segmentation images. These datasets provide one-to-one image alignment, which allows quantitative evaluation through a distance metric such as mean-squared error (MSE) between generated samples and the corresponding ground truth. To facilitate the workings of DDIBs, we additionally employ a color conversion heuristic motivated by optimal transport on image colors (Appendix E.1). Table 3 reports the evaluation results. Surprisingly, DDIBs are able to produce segmentation images that surpass alternative methods in MSE terms; while reverse translations also achieve decent performance.

## 4.4 CLASS-CONDITIONAL IMAGENET TRANSLATION

In this experiment, we apply DDIBs to translation among ImageNet classes. To this end, we leverage the pretrained diffusion models from Dhariwal & Nichol (2021). The authors optimized performance of diffusion models, and end up with a "UNet" (Ho et al., 2020) architecture with particular width, attention and residual configurations. The models are learned on $1,000$ ImageNet classes, each with around $1,000$ training images, and at a variety of resolutions. Our experiments use the model with resolution $256 \times 256$. Moreover, these models incorporate a technique known as *classifier guidance* (Dhariwal & Nichol, 2021), that leverage classifier gradients to steer the sampling process towards arbitrary class labels during image generation. The learned models combined with classifier guidance can be effectively considered as $1,000$ different models. Fig. 4a exhibits select translation samples, where the source images are from ImageNet validation sets. DDIBs are able to create faithful target images that maintain much of the original content such as animal poses, complexions and emotions, while accounting for differences in animal species.

**Multi-Domain Translation**  Given conditional models on the individual domains, DDIBs can be applied to translate between arbitrary pairs of source-target domains, while requiring no additional fine-tuning or adaptation. Fig. 4b displays results of translating a common image of a roaring lion (with class label 291), to various other ImageNet classes. Interestingly, some animals roar, while others stick their tongues out. DDIBs successfully internalize characteristics of distinct animal species, and produce closest animal postures in OT distances to the original shouting lion.

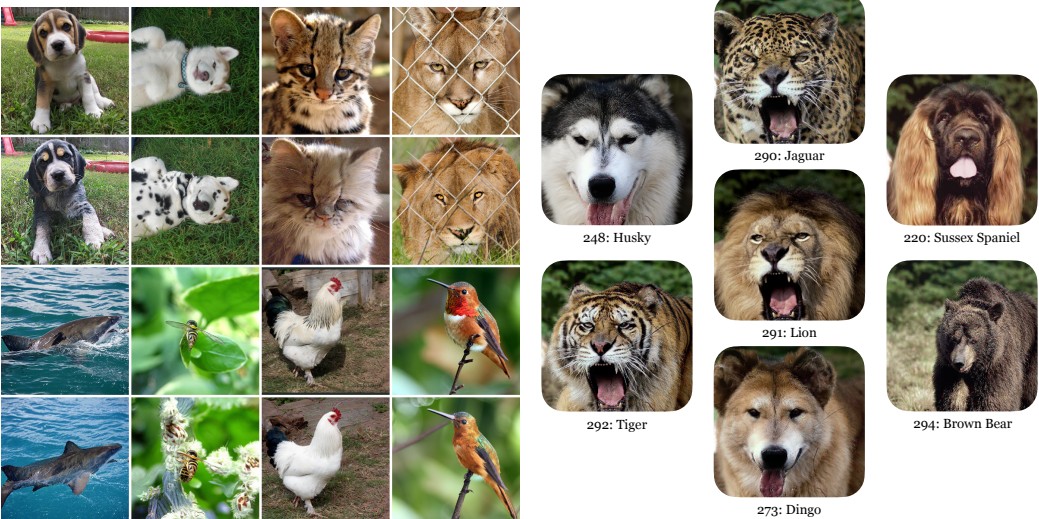

(a) **Conditional ImageNet Translation**: Selected translation samples from various ImageNet classes such as 7: Cock, 94: Hummingbird, 162: Beagle, and 282: Cat.

(b) **Multi-domain translation**: Given the center, source image from class label 291, DDIBs translate it to other animal species, entirely using only a pretrained conditional diffusion model.

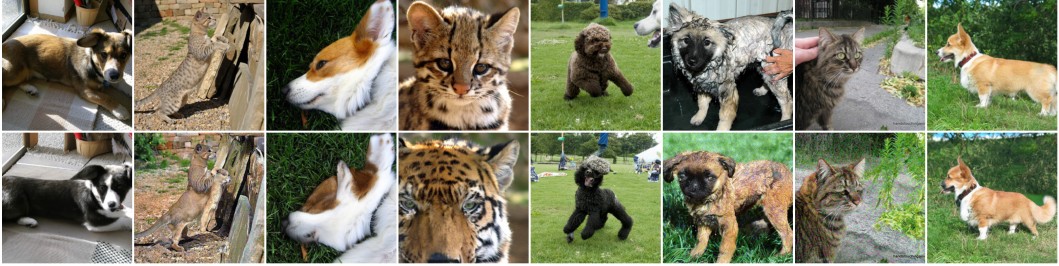

Figure 4: Translation among ImageNet classes.

## 5 RELATED WORKS

**Score-based Diffusion Models**  Originating in thermodynamics (Sohl-Dickstein et al., 2015), diffusion models reverse the dynamics of a noising process to create data samples. The reversal process is understood to implicitly compute scores of the data density at various noise scales, which reveals connections to score-based methods (Song & Ermon, 2019; Nichol & Dhariwal, 2021; Meng et al., 2021b). Diffusion models are applicable to multiple modalities: 3D shapes (Zhou et al., 2021), point cloud (Luo & Hu, 2021), discrete domains (Meng et al., 2022) and function spaces (Lim et al., 2023). They excel in tasks ranging from image editing and composition (Meng et al., 2021a), density estimation (Kingma et al., 2021), to image restoration (Kawar et al., 2022). Seminal works are *denoising diffusion probabilistic models* (DDPMs, Ho et al. (2020)), which parameterized the ELBO objective with Gaussians and, for the first time, synthesized high-quality images with diffusion models; ILVR (Choi et al., 2021), which invented a novel conditional method to direct DDPM generation towards reference images; and *denoising diffusion implicit models* (DDIMs, Song et al. (2020a)), which accelerated DDPM inference via non-Markovian processes. DDIMs can be treated as a first-order numerical solver of a probabilistic ODE, which we use heavily in DDIBs.

**Diffusion Models for Image Translation**  While GANs (Goodfellow et al., 2014; Zhu et al., 2017; Zhao et al., 2020) have been widely adopted in image translation tasks, recent works increasingly leverage diffusion models. For instance, Palette (Saharia et al., 2021) applies a conditional diffusion model to colorization, inpainting, and restoration. DiffuseIT (Kwon & Ye, 2022) utilizes disentangled style and content representation, to perform text- and image-guided style transfer. Lastly, UNIT-DDPM (Sasaki et al., 2021) proposes a novel coupling between domain pairs and trains joint DDPMs for translation. Unlike their joint training, DDIBs apply separate, pretrained diffusion models and leverage geometry of the shared space for translation.

**Optimal Transport for Translation and Generative Modeling**  As it pursues cost-optimal plans to connect image distributions, OT naturally finds applications in image translation. For example, Korotin et al. (2022) capitalizes on the approximation powers of neural networks to compute OT plans between image distributions and perform unpaired translation. By contrast, the entropy-regularized OT variant, Schrödinger Bridges (Section 2), are also commonly used to derive generative models. For instance, De Bortoli et al. (2021) and Vargas et al. (2021) concurrently proposed new numerical procedures that approximate the Iterative Proportional Fitting scheme, to solve SBPs for image generation. Wang et al. (2021) presents a new generative method via entropic interpolation with an SBP. Chen et al. (2021a) discovers equivalence between the likelihood objectives of SBP and score-based models, which lays the theoretical foundations behind DDIBs. Their sequel (Liu et al., 2023) then directly learns the Schrödinger Bridges between image distributions, for applications in image-to-image tasks such as restoration. While DDIBs were not initially designed to mimic Schrödinger Bridges, our analysis reveals their true characterization as solutions to degenerate SBPs.

## 6 CONCLUSIONS

We present Dual Diffusion Implicit Bridges (DDIBs), a new, simplistic image translation method that stems from latest progresses in score-based diffusion models, and is theoretically grounded as Schrödinger Bridges in the image space. DDIBs solve two key problems. First, DDIBs avoid optimization on a coupled loss specific to the given domain pair only. Second, DDIBs better safeguard dataset privacy as they no longer require presence of both datasets during training. Powerful pretrained diffusion models are then integrated into our DDIBs framework, to perform a comprehensive series of experiments that prove DDIBs' practical values in domain translation. Our method is limited in its application to color transfer, as one model is required for each image, which demands significant compute for mass experiments. Rooted in optimal transport, DDIBs translation mimics the mass-moving process which may be problematic at times (Appendix C). Future work may remedy these issues, or extend DDIBs to applications with different dimensions in the source and target domains. As flowing through the concatenated ODEs is time-consuming, improving the translation speed is also a promising direction.

## ACKNOWLEDGEMENTS

We thank Lingxiao Li and Chris Cundy for insightful discussions about the optimal transport properties of DDIBs. We also thank the anonymous reviewers for their constructive comments and feedback. This research was supported by NSF (#1651565), ARO (W911NF-21-1-0125), ONR (N00014-23-1-2159), CZ Biohub, and Stanford HAI.

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

## A Illustration: Privacy-Sensitive Translation

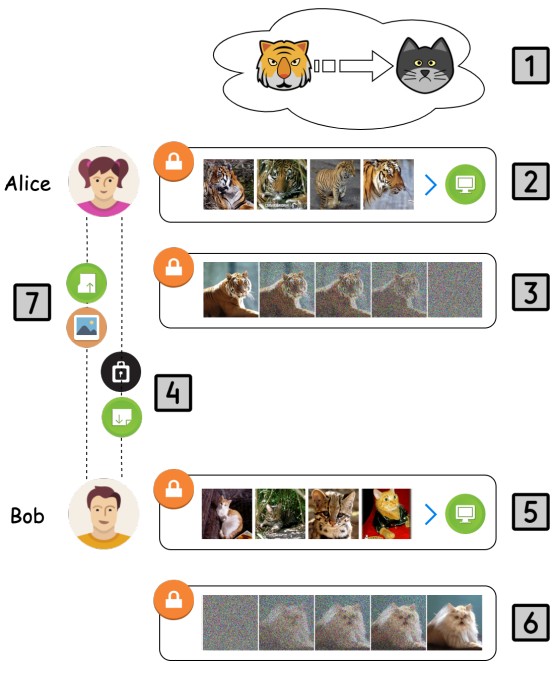

Figure 5

Alice is the owner of the source (tiger) domain, and Bob is the owner of the target (cat) domain. Alice intends to translate tiger images to cat images, but in a privacy-sensitive manner without releasing the source dataset. Bob does not wish to make the cat dataset public, either.

Fig. 5 illustrates the process of privacy-sensitive domain translation. The process contains the following steps, with indexes in the figure.

1. Alice intends to translate tiger images to cat images.
2. Alice trains a diffusion model with the source tiger images.
3. Alice uses the pretrained, tiger diffusion model to convert a source tiger image to its latent code.
4. Alice sends the latent code to Bob.
5. Bob similarly trains a diffusion model on the cat domain.
6. Bob uses the pretrained, cat diffusion model to convert the received latent code to a cat image.
7. Bob then sends the translated image back to Alice.

Clearly, during the translation process, only the latent code and the translated cat image are transmitted via the public channel, while both source and target datasets are private to the two parties. This is a significant advantage of DDIBs over alternate methods, as we enable strong privacy protection of the datasets.

# B  DETAILS OF SGM TRAINING AND DDIM ODE SOLVER

## B.1  TRAINING SCORE NETWORKS

While the description in Section 2 is based on continuous SDEs, actual implementations of diffusion models often sample discrete time steps. Given samples from a data distribution $q(\mathbf{x}_0)$, diffusion models attempt to learn a model distribution $p_\theta(\mathbf{x}_0)$ that approximates $q(\mathbf{x}_0)$, and is easy to sample from. Specifically, diffusion probabilistic models are latent variable models of the form

$$p_\theta(\mathbf{x}_0) = \int p_\theta(\mathbf{x}_{0:T}) \, d\mathbf{x}_{1:T} \,, \text{ where } p_\theta(\mathbf{x}_{0:T}) = p_\theta(\mathbf{x}_T) \prod_{t=1}^{T} p_\theta^{(t)}(\mathbf{x}_{t-1}|\mathbf{x}_t)$$

where $\mathbf{x}_1, \cdots, \mathbf{x}_T$ are latent variables in the same sample space as $\mathbf{x}_0$. The parameters $\theta$ are trained to approximate the data distribution $q(\mathbf{x}_0)$, by maximizing a variational lower bound:

$$\max_\theta \mathbb{E}_{q(\mathbf{x}_0)}[\log p_\theta(\mathbf{x}_0)] \leq \max_\theta \mathbb{E}_{q(\mathbf{x}_0, \mathbf{x}_1, \cdots, \mathbf{x}_T)}[\log p_\theta(\mathbf{x}_{0:T}) - \log q(\mathbf{x}_{1:T}|\mathbf{x}_0)]$$

where $q(\mathbf{x}_{1:T}|\mathbf{x}_0)$ is some inference distribution over the latent variables. It is known that when the conditional distributions are modeled as Gaussians with trainable mean functions and fixed variances, the above objective can be simplified to:

$$L(\epsilon_\theta) := \sum_{t=1}^{T} \mathbb{E}_{\mathbf{x}_0 \sim q(\mathbf{x}_0), \epsilon_t \sim \mathcal{N}(\mathbf{0}, \mathbf{I})} \left[ \left\| \epsilon_\theta^{(t)}(\sqrt{\alpha_t}\mathbf{x}_0 + \sqrt{1-\alpha_t}\epsilon_t) - \epsilon_t \right\|_2^2 \right]$$

The resulting noise prediction functions $\epsilon_\theta^{(t)}$, are equivalent to the score networks $\mathbf{s}_{t,\theta}$ mentioned in Section 2 due to Tweedie's formula (Stein, 1981; Efron, 2011). For details, we refer the reader to Ho et al. (2020); Song et al. (2020a).

## B.2  DDIM ODE SOLVER

With a trained noise prediction model $\epsilon_\theta^{(t)}(\mathbf{x})$, the DDIM iterate between adjacent variables $\mathbf{x}_{t-\Delta t}$ and $\mathbf{x}_t$, considered in Song et al. (2020a), assumes the following form:

$$\frac{\mathbf{x}_{t-\Delta t}}{\sqrt{\alpha_{t-\Delta t}}} = \frac{\mathbf{x}_t}{\sqrt{\alpha_t}} + \left( \sqrt{\frac{1-\alpha_{t-\Delta t}}{\alpha_{t-\Delta t}}} - \sqrt{\frac{1-\alpha_t}{\alpha_t}} \right) \epsilon_\theta^{(t)}(\mathbf{x}_t)$$

In our experiments, we implement the above equation between adjacent diffusion steps. The equation is deterministic, and can be considered as a Euler method over the following ODE:

$$d\bar{\mathbf{x}}(t) = \epsilon_\theta^{(t)} \left( \frac{\bar{\mathbf{x}}(t)}{\sqrt{\sigma^2+1}} \right) d\sigma(t) \tag{9}$$

where we adopt the reparameterization:

$$\sigma(t) = \sqrt{\frac{1-\alpha(t)}{\alpha(t)}}, \quad \bar{\mathbf{x}}(t) = \frac{\mathbf{x}(t)}{\sqrt{\alpha(t)}}$$

Importantly, the ODE in Eq. (9) with the optimal model $\epsilon_\theta^{(t)}(\mathbf{x})$, has an equivalent probability flow ODE corresponding to the "Variance-Exploding" SDE in Song et al. (2020b).

## C  LIMITATIONS OF OPTIMAL TRANSPORT-BASED TRANSLATION

DDIBs contain deterministic bridges between distributions, and are a form of entropy-regularized optimal transport. The learned diffusion models can be effectively considered as a digest or summary of the datasets. While doing translation, they attempt to create images in the target domain, that are closest in optimal transport distances to the source images. Such OT-based process is both an advantage and a limitation of our method.

In ImageNet translation, when the source and target datasets are similar, DDIBs are generally able to identify correct animal postures. For example, we have shouting lions and tigers, because these animals have similar behaviors that are observed in the datasets and then internalized by DDIBs. However, in datasets that are less similar (*e.g.* birds and dogs), DDIBs sometimes fail to produce translation results that retain the postures precisely. We encountered significantly less such cases in AFHQ translation, since the dataset is more standardized and homogeneous.

Fig. 6 illustrates the optimal transport mappings among images as well as some failure cases. Clearly, the translation processes flowing from left to right minimize the Euclidean transportation distances between images. Some of these translated samples may be classified "failure cases" in actual user studies. Such are considered both a feature and a limitation of DDIBs.

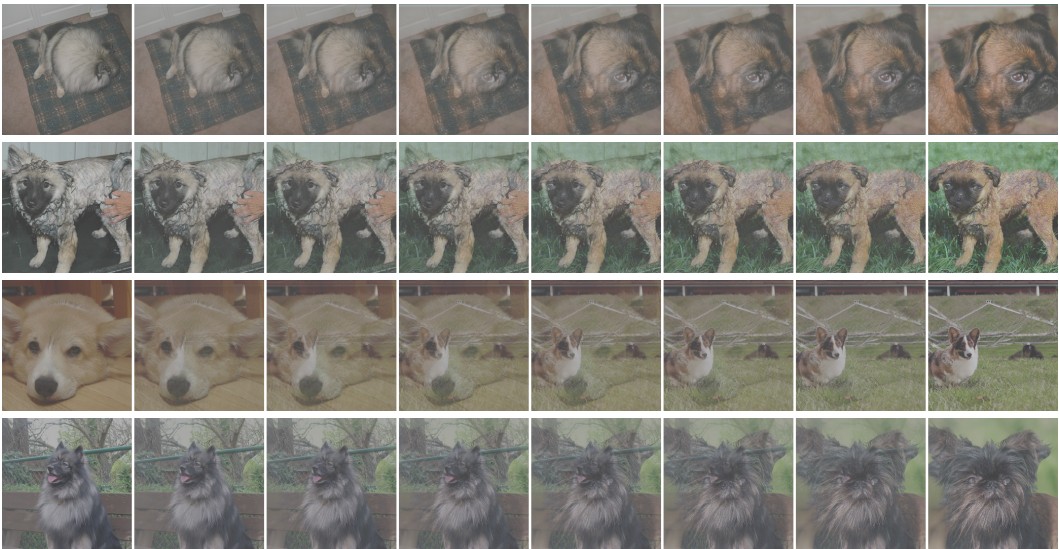

Figure 6: Optimal transport translation processes in DDIBs. *(Leftmost)* Source images. *(Rightmost)* Translated images.

## D  PROOF OF PROPOSITION 3.2

*Proof.* The proof proceeds by substituting the values of $(\mathbf{z}_t, \hat{\mathbf{z}}_t) = (0, g(t)\nabla_{\mathbf{x}} \log p_t(\mathbf{x}))$ into Eq. (6),

$$\mathrm{d}\mathbf{x} = \left[ \mathbf{f}(\mathbf{x}, t) + g(t)\,\mathbf{z} - \frac{1}{2}g(t)(\mathbf{z} + \hat{\mathbf{z}}) \right] \mathrm{d}t \tag{10}$$

$$= \left[ \mathbf{f}(\mathbf{x}, t) - \frac{1}{2}g(t)^2 \nabla_{\mathbf{x}} \log p_t(\mathbf{x}) \right] \mathrm{d}t \tag{11}$$

This is exactly Eq. (2).  □

## E    ADDITIONAL EXPERIMENTAL DETAILS

### E.1    OPTIMAL TRANSPORT IN PAIRED DATASETS

**Color Conversion**    In Fig. 7, a simple examination of the original and segmentation images reveals significant differences in color configurations. In the Maps dataset, while the real, satellite images are composed of dark colors, the segmentation images are light-toned. The same observation applies to other datasets. The shark contrasts in colors intuitively present a large transportation cost, that probably hinders the progress of DDIBs, as we have demonstrated its relationship to OT in Section 3.

To facilitate the workings of DDIBs, we follow a heuristic to transform the colors of the segmentation images. Specifically, on a small subset of the train dataset, we run an OT algorithm to compute a color correspondence that minimizes the color differences in terms of Sinkhorn distances between the real and segmentation images. The segmentation (target) datasets undergo this color conversion before they are fed into a diffusion model for training. During evaluation, when we compute MSEs, the images are converted to the original color space.

**Privacy Protection**    Color conversion requires considering both datasets jointly to compute a color mapping, and seems to betray the original purpose of DDIBs on protection of dataset privacy. We comment that the amount of leaked information is minimal: for example, to compute a color correspondence for the Maps dataset, we sampled only around 1000 pixels from the two datasets, to summarize the color composition information. DDIBs still conserve privacy at large.

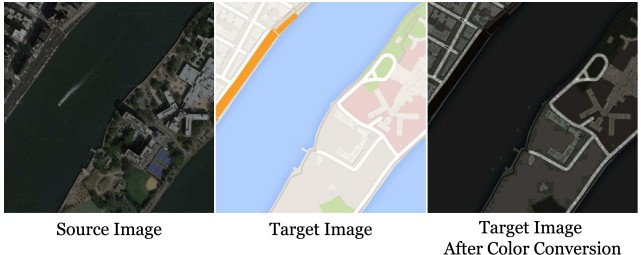


Source Image      Target Image      Target Image
After Color Conversion


Figure 7: **Color Conversion**. In the paired translation tasks, we are given the real and segmentation images. Before training the diffusion models, we first transform the segmentation images to a color palette that is closer to the real images. While evaluating MSEs, we convert the images back to the original colors.

## E.2 EXAMPLE-GUIDED COLOR TRANSFER

We present additional qualitative comparison between DDIBs and common OT methods, in Fig. 8.

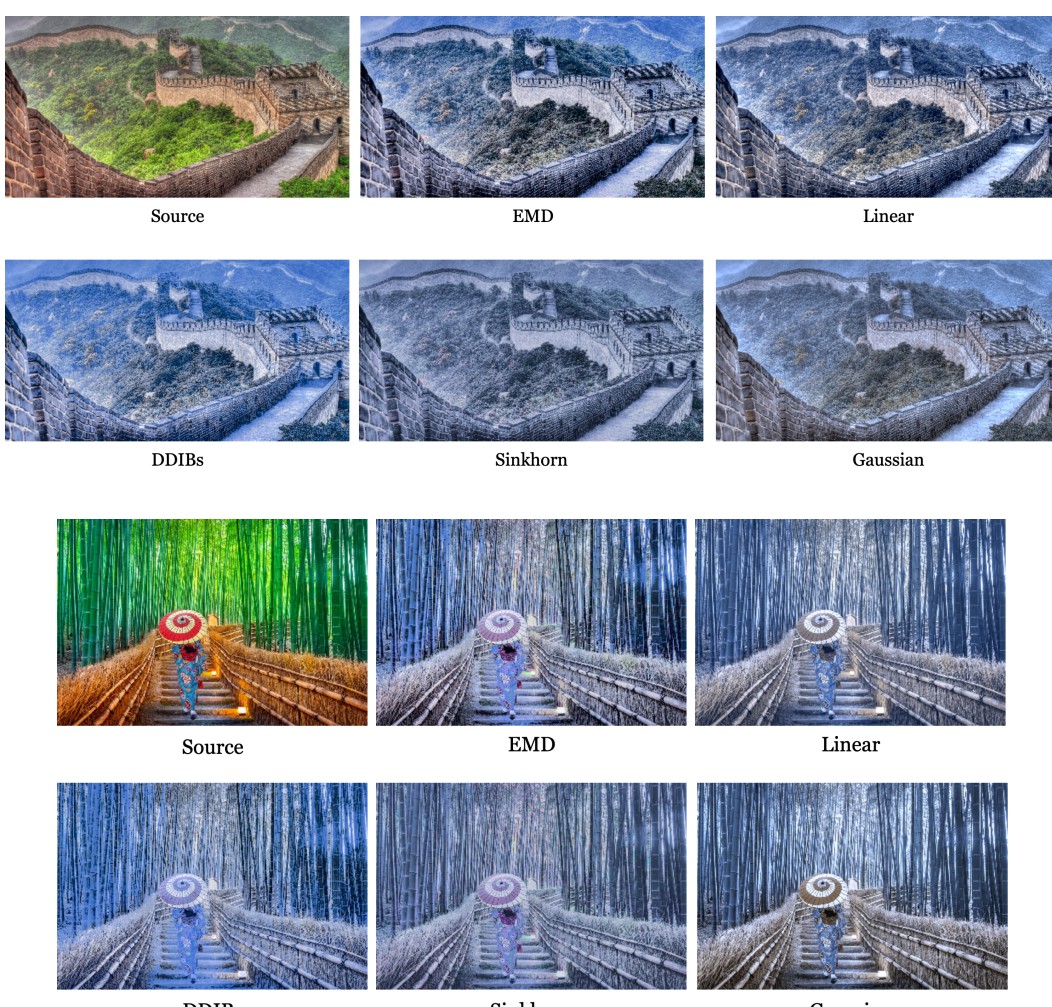

Figure 8: Full color transfer results on example images.

