# OpenReview forum: "Dual Diffusion Implicit Bridges for Image-to-Image Translation"
_ICLR.cc/2023/Conference — ICLR 2023 poster_

### Official Review · Reviewer_2Rpq · 2022-10-21

**Confidence:** 3
**Correctness:** 3
**Technical Novelty And Significance:** 3
**Empirical Novelty And Significance:** 3
**Recommendation:** 5

**Clarity, Quality, Novelty And Reproducibility:**

The paper is overall well-organized. The theoretical analyses are somewhat sufficient. But the innovation of the proposed DDIBs seems unclear.

**Strength And Weaknesses:**

The authors perform a lot of theoretical analyses of the proposed DDIBs from an optimal transport viewpoint. However, the actual implementation of DDIBs remains elusive. The detail comments are as follows.
1. How to train a DDIBs model? It is mentioned that two diffusion models are used in image encoding and decoding. How to train these two diffusion models? What is the loss function?
2. The authors state "Our method is limited in its application to  color transfer, as one model is required for each image, which demands significant compute for mass  experiments." For the color transfer task, why one model is required for each image?
3. Too few comparison methods are used in the experiments. Except for Section 4.2, there is no comparison method in the rest of the experiment. The paper presents only several quantitative experimental results. Moreover, the provided quantitative results is not enough to show the superiority of the proposed method. All this makes the experimental part less convincing.
4. In abstract, the full name is not given when ODE first appears.

**Summary Of The Paper:**

The paper propose an image translation method based on diffusion models for preserving privacy of domain data. Theoretical analyses explain the efficacy of the method. Experimental results demonstrate the utility of the proposed method in a wide variety of translation tasks.

**Summary Of The Review:**

The motivation for this paper is interesting, but the implementation details of the method are unclear and the experiment part is not convincing.

---

> ### Author Response · Authors · 2022-11-11
> **Response to inquiries raised by reviewer 2Rpq**
>
> Q: “How to train a DDIBs model? It is mentioned that two diffusion models are used in image encoding and decoding. How to train these two diffusion models? What is the loss function?”
> - Models in DDIBs are simply diffusion probabilistic models. Training such models relies on an usual variational bound on the negative log likelihood of the dataset.
> - We refer the reviewer to papers such as “Denoising Diffusion Probabilistic Models” or surveys like “Diffusion Models: A Comprehensive Survey of Methods and Applications”, for details about training diffusion models and their elementary formulation.
> - We also included an additional section in the appendix to describe the procedure for training diffusion models.
>
> Q: “The authors state "Our method is limited in its application to color transfer, as one model is required for each image, which demands significant compute for mass experiments." For the color transfer task, why one model is required for each image?”
> - For each image, we train a diffusion model on its RGB color space. Translation of image colors then follows the usual procedure, by first translating the colors of the source image to the latent space, and then to the target space.
>
> Q: “In abstract, the full name is not given when ODE first appears.”
> - In the revised version, we have edited the abstract to make the ODE term clearer.

---

### Official Review · Reviewer_Ztwc · 2022-10-23

**Confidence:** 4
**Correctness:** 4
**Technical Novelty And Significance:** 2
**Empirical Novelty And Significance:** 2
**Recommendation:** 5

**Clarity, Quality, Novelty And Reproducibility:**

Clarity: High

Quality: High and easy to follow

Novelty: Limited

Reproducibility: High

**Strength And Weaknesses:**

Stength
1. The paper is clearly written and easy to follow.
2. The experiments showed inspiring image translation scenarios (image corloring, animal species translation and etc.)


Weakness
1. The proposed method is straightfoward application of diffusion model of limited novelty (e.g. using foward difussion (DDIM) to extract image latent for reconstuction has been explored before)
2. The results on translation varies a lot depending on the source domain and target domain (e.g. the good examples of species translation are among the close related species while others do not preserve the layout and etc., can we do some evaluations on this ? for example how hard is it to translate cars to ships ? )
3. People have observed that for diffusion models, certain sampling decisions result in similarly structured pictuers (e.g. random seed), are such decisions already considered ?

**Summary Of The Paper:**

This paper proposes to do image translation by first converting source image with forward diffusion, and then run reverse diffusion in the target domain. Authors draw connection of this method with Schrodinger Bridge problem, demonstrated qualitative results on 2D synthetic datasets, color transfer and validated cycle consistency on image translations. For quantitative evaluations they compared pixe-wise MSE on paried datasets Facades and Maps and showed better results on one-direction translation but worse on the other direction.

**Summary Of The Review:**

The paper applies diffusion model to image translation and showed some good translation examples. The method is very straightfoward though and quantitative results are a bit mixing : A->B translation shows superior peromance but not B->A. Even though the results are interesting to see but the analysis could go a bit deeper: for example the translated image does not repsect preservation of details, translation is only good between very closely related domains and etc.

---

> ### Author Response · Authors · 2022-11-11
> **Response to inquiries raised by reviewer Ztwc**
>
> Q: “The proposed method is straightfoward application of diffusion model of limited novelty (e.g. using foward difussion (DDIM) to extract image latent for reconstuction has been explored before)”
> - Indeed, image reconstruction with diffusion models has been exploited in similar works.
> - However, we claim that our method contains sufficient novelty, due to two critical differences from other works.
> - First, as you suggested, image reconstruction has been done with DDIMs. However, what we are doing is image-to-image translation, not reconstruction. We are using two different (separate, or independent) diffusion models whereas image reconstruction only requires one. Thus, our finding that diffusion models are well-suited for image-to-image translation without additional task-specific training (like in GANs) is novel.
> - Second, apart from the diffusion process, our work additionally establishes close connections to optimal transport processes between the two domains. This is a theoretical advantage and explanation of our translation method, unseen in prior image reconstruction works.
>
> Q: “The results on translation varies a lot depending on the source domain and target domain.”
> - Indeed, we also observed that translation between species or categories that are similar to each other, is more faithful and successful; while translation between vastly different species or between, say, ships and cars, often results in failures and unsatisfactory images.
> - However, this is perhaps normal and expected of an image translation method; because in practice and in most use cases, it is more meaningful and applicable to translate between similar classes of objects (e.g. between different cat breeds), than between dissimilar categories. Even in the CycleGAN paper, the authors have mentioned that “our model does not work well when a test image looks unusual compared to training images”, further emphasizing the importance of the choice of domain in this ill-posed problem.
>
>
> Q: “People have observed that for diffusion models, certain sampling decisions result in similarly structured pictures (e.g. random seed), are such decisions already considered ?”
> - Different from other diffusion works, the sampling in DDIBs is deterministic and is not affected by external factors such as random seeds. The mappings between images and latents are bijective (up to negligible discretization errors), and the DDIBs formulation doesn’t need to make sampling decisions unlike in prior works.

---

### Official Review · Reviewer_FZcm · 2022-10-24

**Confidence:** 5
**Correctness:** 4
**Technical Novelty And Significance:** 4
**Empirical Novelty And Significance:** 4
**Recommendation:** 10

**Clarity, Quality, Novelty And Reproducibility:**

The paper is well-written with all concepts clearly introduced. And the proposed method is novel, cause I haven't seen any similar contributions in diffusion-based methods. Additionally, I am very confident to say this work is reproducible, cause I have run the codes.

**Details Of Ethics Concerns:**

Since it works on public and synthetic datasets, I don't have ethics concerns.

**Strength And Weaknesses:**

Strength:
* The paper is well-written and well-organized.
* The idea of treating the modified DDIM as Schrodinger Bridges is novel.
*  The work is easy to follow. I have run the codes and found it could get stable results on small datasets, which can not be achieved by the SOTA general image-to-image translation model, Palette [A].
To be honest, I have utilized it as a comparison baseline in my work now.

Weakness:
* The comparisons included in experiments are not sufficient. Many baselines, such as MUNIT and Stargan should be included. Furthermore, it lacks quantitative comparisons.

[A] Saharia, Chitwan, et al. "Palette: Image-to-image diffusion models." ACM SIGGRAPH 2022 Conference Proceedings. 2022.

**Summary Of The Paper:**

This paper proposes an unpaired image-to-image translation method, which consists of two modified Denoising Diffusion Implicit
Models (DDIMs). The authors s theoretically demonstrate the modified DDIM as Schrodinger Bridges and name it DDIB. Experimental studies on both paired image-to-image translation and unpaired image-to-image translation are included. As far as my knowledge goes, the proposed work is the first diffusion model to successfully achieve the unpaired image-to-image translation with open resources (some unpublished works, such as UNIT-DDPM, do not have released codes).

**Summary Of The Review:**

The authors theoretically demonstrate their proposed modified DDIM as Schrodinger Bridges. Experimental studies support their claimed contributions. As far as my knowledge goes, the proposed work is the first diffusion model to successfully achieve the unpaired image-to-image translation with open resources. Although this paper has some drawbacks in the comparison parts, it is novel and reproducible. I am very happy to see this impactful work published since it has been released for a while.

---

> ### Author Response · Authors · 2022-11-11
> **Response to inquiries raised by reviewer FZcm**
>
> We would like to acknowledge the good rating and comments reported by this reviewer. We also thank the reviewer for having already experimented with the codebase and confirming its reproducibility. Thank you!

---

### Official Review · Reviewer_iEhQ · 2022-10-26

**Confidence:** 3
**Correctness:** 3
**Technical Novelty And Significance:** 3
**Empirical Novelty And Significance:** 3
**Recommendation:** 6

**Clarity, Quality, Novelty And Reproducibility:**

Clarity: The paper is nicely written.

Novelty: It is rather easy to conceive the idea of using the latent diffusion output as bridge. But putting it under a solid theorectic foundation is novel, offering the necessary justifcation.

Reproducibility: It is not impossible to reproduce the work. Supplying the codes will make it much easier.

Quality: A decent paper with theorectic contribution and abundant translation results.

**Strength And Weaknesses:**

Strength: The ODE-based theoretic foundation is solid, which justifies the bridge idea based on the latent space. In terms of experiments, there are abundant results presented in the paper as well as in the supplementary materials. The synthetic image translation task nicely demonstrates the merit of this work.

Weaknesses:

While the author discusses the privacy protection of medical images, they lack corresponding experiments to confirm the translation performances on medical images.
While natural image patterns are usually varying and contain different foreground&background contexts even within one class, medical image translation focuses the detail preservation. Therefore, it is not clear if such a separately training method is capable to handle the task.

Considering the privacy protection problem, we can employ two well-pretrained auto-encoders, named A&B, for both source&target domains(denoted S and T) to obtain latent encodings A(S) and B(T), which mapping retains domain information mostly. Then we can utilize other image-to-image translation models (GAN-based or normalizing flow-based) to jointly train a model on latent encodings and protect data privacy. I am curious about the practical differences and performance differences between the DDIB and such a direct manner. Can the authors provide a detailed discussion?

**Summary Of The Paper:**

The paper proposes the novel DDIB for the image-to-image translation task, which diffuses&reversely diffuses in both source and target domains separately. This method only needs to independently train both diffusion processes and better safeguard dataset privacy. The paper also presents an ODE-based theoretic foundation to describe the latent space based translation as an entropy-regularized optimal transport. Experiments are conducted on several image translation tasks using synthetic and high-resolution image datasets.

**Summary Of The Review:**

In sum, it is a decent paper with strengths in its theorectic foundings and extensive experiments. The synthetic image translation results look really impressive. But it lacks the results on say medical images, where privacy concern is real. I believe that the paper opens a new direction of applying the DDPM method to image translation across different domains.

---

> ### Author Response · Authors · 2022-11-11
> **Response to inquiries raised by iEhQ**
>
> Q: “They lack corresponding experiments to confirm the translation performances on medical images.”
> - The example on medical images is given to illustrate a potential use case of our method on privacy-sensitive translation. Unfortunately, mainstream image translation datasets are not specifically designed for medical applications; we (and other ML or medical practitioners) don’t have a medical dataset to test our translation algorithm.
>
> Q: “Considering the privacy protection problem, we can employ two well-pretrained auto-encoders, named A&B, for both source&target domains(denoted S and T) to obtain latent encodings A(S) and B(T), which mapping retains domain information mostly. Then we can utilize other image-to-image translation models (GAN-based or normalizing flow-based) to jointly train a model on latent encodings and protect data privacy. I am curious about the practical differences and performance differences between the DDIB and such a direct manner. Can the authors provide a detailed discussion?”
> - We would like to thank the reviewer for brainstorming the novel translation algorithm based on autoencoders. We think that this is potentially a viable image translation method, and indeed, the method also enables privacy protection of the datasets.
> - However, our method still has the following advantages.
> - First, DDIBs are more straightforward, and naturally arise from diffusion models. In fact, we can consider the domain-specific diffusion models as natural, separate “autoencoders” on the two domains. In comparison, the method proposed by the reviewer requires an additional translation procedure between the two latent spaces, which will add additional, perhaps unnecessary complexity.
> - Second, the unpaired domain translation task is ill-defined in principle. Specifically, for unpaired translation, given a cycle-consistent model, one can permute the translated image pairs such that they are still cycle-consistent: for example, given cycle-consistent translations A → B → A and C → D → C, the permuted pairs A → D → A and C → B → C are also cycle-consistent. In practice, successful domain translation without paired data would rely on certain priors. In image-to-image translation (such as CycleGAN), the convolutional model architecture and the optimization process would encourage solutions that are more reasonable. However, the same is hard to say for the latent space, where we do not have image-based prior knowledge and this is an advantage of DDIBs.
> - Last but not least, we want to emphasize that DDIBs are complementary to the approach that the reviewer proposed, since we also encode the images into suitable latent variables. DDIBs can be considered a special case of the proposed method: they are simply two "autoencoders", and an identity mapping over the latent space. While using an identity mapping has already shown promising results, we can also perform joint training over these latent embeddings, which may further improve performance.
>
> Q: “It is not impossible to reproduce the work. Supplying the codes will make it much easier.”
> - We plan to include a link to our codebase in the final revision of this paper. For now, we have uploaded our code repository to the updated supplementary materials of our submission.
> - Moreover, reviewer FZcm has already experimented with our codebase and reported good experimental performance. This supports the reproducibility of our work.

---

### Author Response · Authors · 2022-11-11
**Overview Comment**

We would like to thank the effort from the reviewers to read and provide valuable comments about our work.

To address reviewers’ concerns, we have made the following major changes:
1. We included a repository containing our source code, in the supplementary materials.
2. We uploaded a new revision of our paper, and highlighted the parts relevant to inquiries raised by the reviewers, in red. We also made a number of other miscellaneous edits to our paper.

In addition, we provide individual comments to address the feedback raised by the reviewers.

Let us know if you have additional concerns / comments!

---

### Public Comment · ~James_Thornton1 · 2023-03-07
**Connection between SBP and SGM was established by Bortoli et al.**

It is stated "Chen et al. (2021a) establishes connections between SGMs and SBPs".

Chen et al 2021 [2] was **~5 months after** Bortoli 2021[1], which is the first work to connect SGM and SBP. The dates may be verified on the links below.

[1] Diffusion Schrödinger Bridge with Applications to Score-Based Generative Modeling, 2021, De Bortoli et al https://arxiv.org/abs/2106.01357 \
[2] Likelihood Training of Schrödinger Bridge using Forward-Backward SDEs Theory, Chen et al 2021 https://arxiv.org/abs/2110.11291

---

> ### Author Response · Authors · 2023-03-07
> **Thank you for the comment!**
>
> Thank you for pointing that out! We will update our paper.

---

### Decision · Program_Chairs · 2023-01-20

**Decision:**

Accept: poster

**Justification For Why Not Higher Score:**

The work is not novel as it builds on diffusion models but it provides a nice application in cycle consistent generation.

**Justification For Why Not Lower Score:**

The paper presents a good idea for cycle consistent generation and it is backed up with thorough experimentation and is reproducible.

**Metareview: Summary, Strengths And Weaknesses:**

The paper proposes to use two independently trained diffusion models on different image domains. The source diffusion models encodes the the source image to a latent domain and then it is decoded to the target domain using the target diffusion model.
The paper shows that the continuous diffusion resulting from applying successively these diffusion models is cycle consistent.
The code was provided during  the rebuttal period, making the work reproducible. The cycle consistency of the proposed method being a feature by design of the method makes it an appealing method for translation between unpaired domains.



**Note From Pc:**

if the above contains the word "oral" or "spotlight" please see: "oral" presentation means -> notable-top-5% and "spotlight" means -> notable-top-25%. As stated in our emails, we are disassociating presentation type from AC recommendations

**Summary Of Ac-Reviewer Meeting:**

AC tried to reach out multiple times to reviewers and tried to organize a call with the reviewers without responses from reviewers.  The AC decision was made based on their own assessment of the work.